# OssaNMA: An R package for using information from network meta-analyses to optimize the power and sample allocation of a new two-arm trial

**Fangshu Ye**[1], **Chong Wang**[1,2]*, **Annette M. O'Connor**[2,3]

**1** Department of Statistics, Iowa State University, Ames, Iowa, United States of America, **2** Department of Veterinary Diagnostics and Production Animal Medicine, Iowa State University, Ames, Iowa, United States of America, **3** Department of Large Animal Clinical Sciences, Michigan State University, East Lansing, Michigan, United States of America

* chwang@iastate.edu

**Data Availability Statement:** We provide the data and R code we used in this paper in https://github.com/fangshuye/OssaNMA_replication_files.

## Abstract

Randomized clinical trials (RCTs) are designed for measuring the effectiveness of the treatments and testing a hypothesis regarding the relative effect between two or more treatments. Trial designers are often interested in maximizing power when the total sample size is fixed or minimizing the required total sample size to reach a pre-specified power. One approach to maximizing power proposed by previous researchers is to leverage prior evidence using meta-analysis (NMA) to inform the sample size determination of a new trial. For example, researchers may be interested in designing a two-arm trial comparing treatments A and B which are already in the existing trial network but do not have any direct comparison. The researchers' intention is to incorporate the result into an existing network for meta-analysis. Here we develop formulas to address these options and use simulations to validate our formula and evaluate the performance of different analysis methods in terms of power. We also implement our proposed method into the R package OssaNMA and publish an R Shiny app for the convenience of the application. The goal of the package is to enable researchers to readily adopt the proposed approach which can improve the power of an RCT and is therefore resource-saving. In the R Shiny app, We also provide the option to include the cost of each treatment which would enable researchers to compare the total treatment cost associated with each design and analysis approach. Further, we explore the effect of allocation to treatment group on study power when the *a priori* plan is to incorporate the new trial result into an existing network for meta-analysis.

## Introduction

Randomized clinical trials (RCTs) are designed for measuring the effectiveness of the treatments and testing a hypothesis regarding the relative effect between two or more treatments. Some important aspects of trial design include calculating the required sample size to achieve the desired level of power and optimizing the allocation of the total sample size to the

**Funding:** The author(s) received no specific funding for this work.

**Competing interests:** The authors have declared that no competing interests exist.

treatment groups to maximize the power. However, it can be difficult for a single trial to achieve a sufficient level of power with resources limited or constrained by the facility. To improve the power of trials to detect meaningful difference in effect, evidence synthesis can be used, such that, there is an *a priori* plan is to incorporate the new trial result into an existing trial network for meta-analysis (NMA). Researchers have proposed methods to plan a randomized clinical trial specifically to update a pairwise meta-analysis [1, 2] and others have recently extended to NMA [3–5]. There have been a few published applications of the use of NMA to inform the sample size [6, 7] of future trials.

Determining the sample size is an important part of designing RCTs to address a specific scientific question with a pre-specified power. Sutton *et al.* (2007) [1] described a simulation approach to sample-size estimation for a new trial based on the result of an updated meta-analysis. Roloff *et al.* (2012) [2] proposed a method based on the conditional power of a meta-analysis to aid the planning of future trials that avoids the need for simulation, where conditional power is defined as the power to detect a specified overall mean effect size given the existing evidence. Nikolakopoulou *et al.* (2014) [3] provided recommendations for further clinical trials using conditional power of NMA. When compared to the optimal sample size obtained from traditional methods of sample size calculations that use estimates of effect and variance, the estimated optimal sample size of the conditional power method was much smaller. The focus of the work from [3] was to design a series of future two-arm clinical trials with the optimal total sample size for each trial to achieve a pre-specified power. However, designing a series of future trials at the same time is rare in practice. A more common question is the design of a single trial of interest to the researcher with the goal of having the most powerful approach to comparing two treatments. To illustrate the problems above, we concentrate on a special future two-arm study with a binary outcome, which has two arms already in the existing trial network but does not have any direct comparison. Under that condition, we propose a method to calculate the optimal sample size and allocation for each treatment with a fixed total sample size or a pre-specified power.

In addition to determining the optimal sample size for each treatment in a future two-arm trial, another important question is how to make the implementation of the method easier for the trial planners. To use these methods proposed, the results of network meta-analysis of the existing evidence are required. Furthermore, the optimization problems that need to be solved do not have closed-form solutions for sample sizes. Therefore proposed methods require the end-users to conduct a network meta-analysis and solve a non-linear optimization problem. Although packages and code for each aspect of the process are available, these can be difficult to use and be combined, particularly for non-statisticians, which are often the investigators who conduct RCTs. Therefore, we aim to build two tools to make our proposed method accessible and flexible: the first tool is an R [8] package OssaNMA, which calculates the sample size for each treatment group for the new two-arm trial. The second tool is an R Shiny [9] app, which is a user-friendly web-based application of the OssaNMA that gives investigators a web-based interface to obtain the optimal sample size for their future two-arm trial.

To summarize, we have three aims. First, we aim to provide the optimal sample size for each treatment in a future two-arm trial with fixed total sample size. We document the increase in power associated with pre-planning to include the new trial in a network meta-analysis. We also investigate the magnitude of the difference in power between even allocation and the optimal allocation to group, based on the existing evidence. Second, because researchers may not have a fixed sample size, but rather a pre-specified power, we also calculate the optimal sample size for each treatment in a future two-arm trial in order to achieve a pre-specified level of power, based on the existing evidence as well. Finally, we would introduce our R package OssaNMA and our R Shiny app, which are designed to facilitate the calculation of the optimal sample size mentioned in our first two aims.

We organize the rest of the paper as follows. In Materials and methods, we briefly overview the fixed effect network meta-analysis and derive expressions for the optimization problems to solve in order to obtain the optimal sample size or power under different scenarios. In Application and simulation, we report the application and simulation. In Results, we present the simulation results in different scenarios. In Application tool: R Shiny webpage, we provide a detailed tour of the R Shiny webpage and illustrate its use in a 'real-world' setting. In The package, we introduce OssaNMA package. An empirical illustration of applying the package is provided in Empirical illustrations. In Discussion and Conclusion, we discuss the potential implications and limitations of our proposed method and tool.

## Materials and methods

Suppose a new two-arm trial is to be conducted comparing treatment A and B. The researchers can be in two circumstances. First, they may have a fixed total sample size due to financial or facilities limitations. In such a circumstance, the goal is to maximize the power of the new trial when the total sample size is fixed. Alternatively, some researchers may want to reach a pre-specified power and therefore the goal is to minimize the required total sample size.

The organization of the five subsections is structured as follows: in the first subsection, we discuss the traditional approach to sample size determination and optimal sample size allocation with a fixed total sample size without an existing NMA. In the second subsection, we provide a brief introduction to the fixed-effect network meta-analysis model, which provides the theoretical basis for the next subsection, in which we derive the formula for the variance of the estimated relative effect size when we analyze the new trial with the existing network, and then utilize this variance estimate to determine the optimal allocation of study subjects to groups under a fixed total sample size in the final subsection. Fig 1 illustrates the key formulas for calculating the optimal sample size allocation with a fixed total sample size.

In addition to solving the optimization problem related to maximizing the power with a fixed total sample size, a method to minimize the total sample size required when the goal is to achieve a pre-specified power is also vital. We provide he formula for the power when the new trial is analyzed with or without the existing trial network by utilizing the variance estimate. Given the formula of power, we document the approach to obtaining the optimal sample size when the goal is to reach a pre-specified power. Fig 2 summarises the key parts in for calculating the optimal required sample size and how to allocate it with a pre-defined power.

### Traditional methods for a two-arm trial

Suppose the total sample size, $n$, for the new two-arm trial is fixed. We want to obtain the best sample allocation strategy by minimizing the standard error of the estimated effect size. Let the number of samples assigned to treatment A and B be $n_A$ and $n_B$, respectively, under the condition $n_A + n_B = n$.

Suppose the outcome is binary such as a disease event. To simplify the notation, we use $r_i$ to denote the number of events in the $i$-th treatment, $\pi_i$ to denote the probability of an event occurring in treatment $i$, $i \in A, B$.

The logistic regression model for this problem is

$$r_i \sim \text{Binomial}(n_i, \pi_i),$$

$$\log\left(\frac{\pi_i}{1 - \pi_i}\right) = \beta_1 + \beta_2 I_{(i=B)}.$$

Let $\mu_{AB,\text{trial}}$ be the relative effect size of treatment B to A, and $\hat{\mu}_{AB,\text{trial}}$ be the estimate of $\mu_{AB,\text{trial}}$.

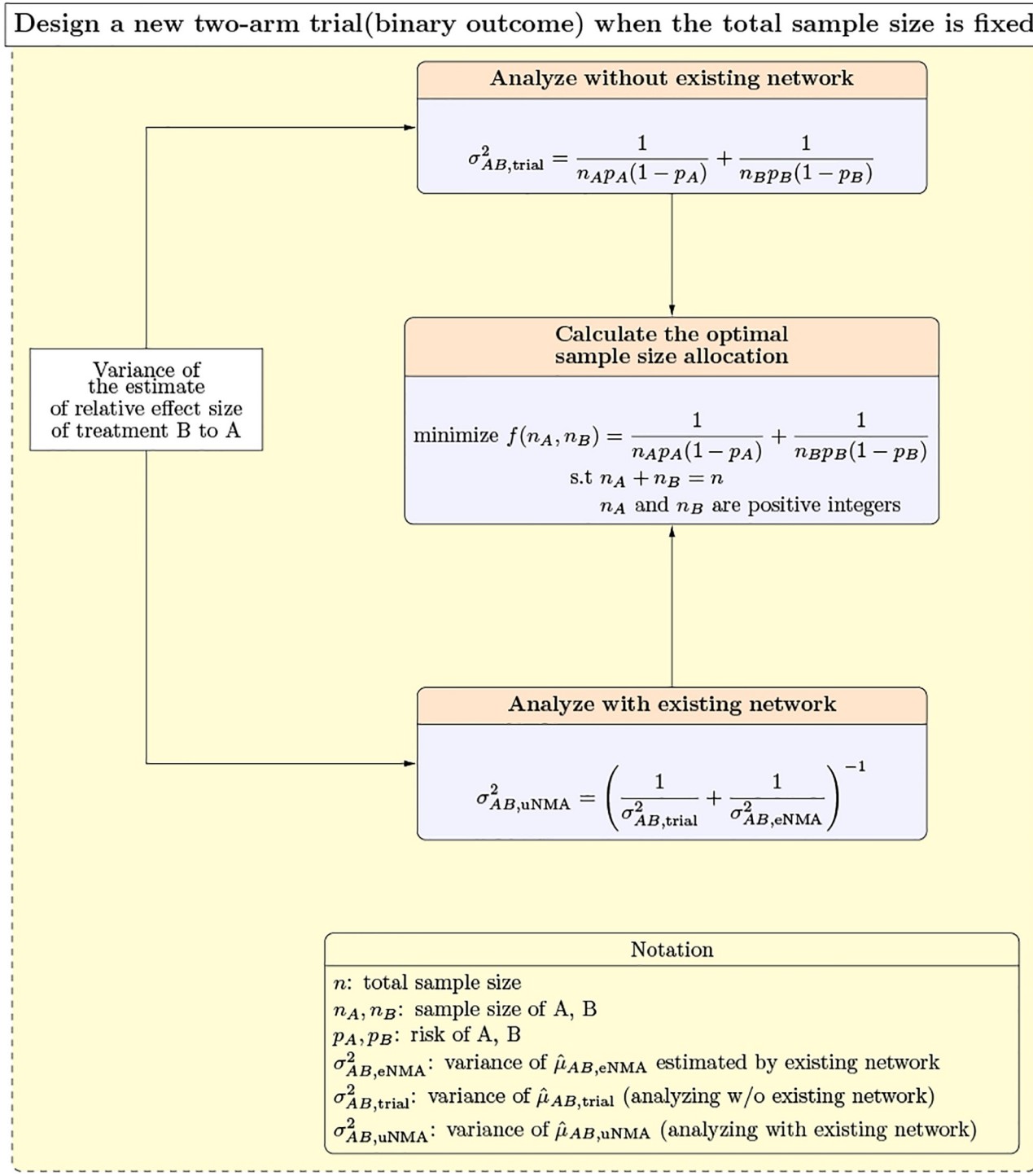

**Fig 1. Variance formulas and the optimization problem in a new two-arm trial when our goal is to get maximum power with a fixed total sample size.**

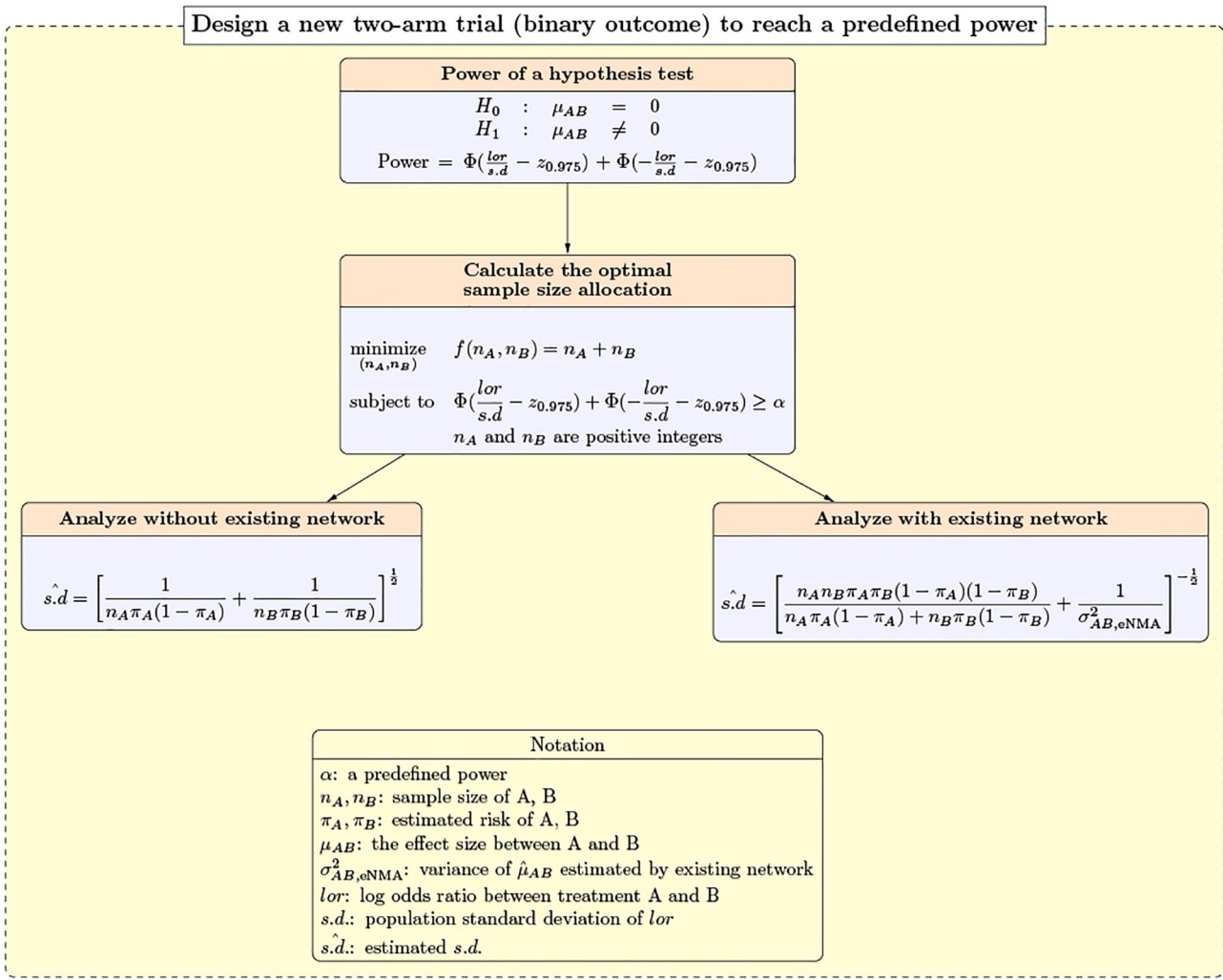

**Fig 2. Power formulas and the optimization problem in a new two-arm trial when our goal is to minimize total sample size with a pre-specified power.**

We have

$$\hat{\mu}_{AB,\text{trial}} = \hat{\beta}_2.$$

Let $\sigma^2_{AB,\text{trial}}$ be the variance of $\hat{\mu}_{AB,\text{trial}}$. Then

$$\sigma^2_{AB,\text{trial}} = \text{Var}(\hat{\beta}_2) = \frac{1}{n_A p_A (1 - p_A)} + \frac{1}{n_B p_B (1 - p_B)}. \tag{1}$$

where $p_i = r_i/n_i$ is the estimated $\pi_i$ for $i \in A, B$. Hence, the optimization problem is

$$\text{minimize } f(n_A, n_B) = \frac{1}{n_A p_A (1 - p_A)} + \frac{1}{n_B p_B (1 - p_B)}. \tag{2}$$

$$\text{s.t } n_A + n_B = n$$

$$n_A \text{ and } n_B \text{ are positive integers.}$$

Suppose the outcome is continuous such as a production metric like average daily gain. Let $y_s$ denote the outcome in the $s$-th animal and $Trt_s$ denote the treatment that the $s$-th animal received. The linear regression model for $s = 1, \cdots, n$ is

$$y_s = \beta_1 + \beta_2 I_{(Trt_s=B)} + \epsilon_s,$$

where $\epsilon_s$ is iid with N(0,$\sigma^2$).

Using the same notation as the binary case, we have

$$\hat{\mu}_{AB,\text{trial}} = \hat{\beta}_2.$$

and

$$Var(\hat{\beta}_2) = \sigma^2 \times \left( \frac{1}{n_A} + \frac{1}{n_B} \right).$$

To minimize $Var(\hat{\beta}_2)$, the optimal allocation would be $n_A = n_B = \frac{n}{2}$ if $\sigma^2$ is assumed known.

In the case of a binary outcome (Eq 2), it is straightforward to tell that the optimal allocation approach would be uneven unless $p_A = p_B$ whereas even allocation is always the optimal in the continuous case.

## Fixed-effect network meta-analysis (NMA) model

Consider a network of $T$ treatments with $I$ studies, with the $i$th study having $n_i$ arms. The treatments in each study form a subset of $T$ treatments. There are $\binom{T}{2}$ pairwise comparisons, each associated with a relative treatment effect. We denote all the treatment effects by a vector $\boldsymbol{\mu_f}$. A critical aspect of our network meta-analysis model is the consistency assumption [10–12], which allows pairwise comparisons between the treatments of interest to be estimated as functions of the basic parameters. The basic parameters are the relative effects of all treatments to the baseline treatment. For example, there are three treatments A, B and C in the network and A is chosen to be the baseline treatment. The consistency assumption is written as:

$$\mu_{BC} = \mu_{AC} - \mu_{AB}.$$

A fixed effect NMA assumes that there is only one effect size underlying the trials for each comparison. It follows that all of the differences in the observed effect sizes of a pairwise comparison are due to random variation (sampling error) [13]. For example, let $y_{j,AB}$ be the observed effect size of treatment B to A in the $j$-th studies contains treatments A and B. Then,

$$y_{j,AB} = \mu_{AB} + \epsilon_j,$$

where $\epsilon_j$ is the within-study error.

In a network of $T$ treatments with $I$ studies, assume there are $n_i$ treatments in study $i$. Let $\boldsymbol{y_i}$ denote the observed effect size for the $i$th study, $\boldsymbol{y_i} = (y_{i,1}, \ldots, y_{i,ni}-1)^T$ and $\boldsymbol{y} = (\boldsymbol{y_1}, \ldots, \boldsymbol{y_I})$. Let $\boldsymbol{\mu_i}$ be the vector of the true effect sizes of study $i$ and $\boldsymbol{\mu_s} = (\boldsymbol{\mu_1}, \ldots, \boldsymbol{\mu_I})$. Then we have

$$\boldsymbol{y_i} = \boldsymbol{\mu_i} + \epsilon_i, \quad i = 1, \ldots, n_i$$

where $\epsilon_i$ represents the vector of errors of study $i$. $\epsilon_i$ is assumed to be normally distributed and

independent across studies and its covariance is $\text{cov}(\epsilon_i) = S_i$. $S_i$ is a diagonal matrix of size $(n_i - 1) \times (n_i - 1)$ and is a scalar if study $i$ only has two arms. The distribution of $\boldsymbol{y}$ is

$$\boldsymbol{y} \sim \text{MVN}(\boldsymbol{\mu_s}, \boldsymbol{S}),$$

where $\boldsymbol{S}$ is a block diagonal matrix with each block $\boldsymbol{S_i}$, $i = 1, \ldots, I$. Let $\boldsymbol{\mu}$ be a sub-vector of $\boldsymbol{\mu_s}$ of length $T - 1$ that involves the relative effect sizes of all treatments to a baseline. We name the comparisons in $\boldsymbol{\mu}$ basic comparisons. Apparently, $\boldsymbol{\mu_s}$ is a linear combination of $\boldsymbol{\mu}$ and can be written as $\boldsymbol{\mu_s} = \boldsymbol{X_\mu}$, where $\boldsymbol{X}$ is the design matrix of size $\sum_{j=1}^{j} n_j \times (T - 1)$. Each row of $\boldsymbol{X}$ corresponds to a treatment comparison in a study and each column represents a basic comparison. There are only two possible situations in each row of $\boldsymbol{X}$. First situation is that there is only one element of 1 and other elements are 0, when this comparison is a basic comparison. The second situation is that 1 and -1 occur in one row and others are 0, when this comparison can be written as a function of the basic comparisons, which is contingent on consistency assumption. The distribution of $\boldsymbol{y}$ is then

$$\boldsymbol{y} \sim \text{MVN}(\boldsymbol{X\mu}, \boldsymbol{S}).$$

The maximum likelihood estimate of $\boldsymbol{\mu}$ and its variance are

$$\hat{\boldsymbol{\mu}} = (\boldsymbol{X}^T \boldsymbol{S}^{-1} \boldsymbol{X})^{-1} \boldsymbol{X}^T \boldsymbol{S}^{-1} \boldsymbol{y},$$

$$\text{Var}(\hat{\boldsymbol{\mu}}) = (\boldsymbol{X}^T \boldsymbol{S}^{-1} \boldsymbol{X})^{-1}. \tag{3}$$

## Analysis of the new trial with the existing trial network

Suppose the researchers aim to conduct a new two-arm trial comparing treatment A and B, where A and B are treatments included in the existing network but do not have direct comparison i.e. no trial comparing A and B is available. Let $\mu_{AB,\text{uNMA}}$ be the relative effect size of treatment B to A, and $\hat{\mu}_{AB,\text{uNMA}}$ be the estimate of $\mu_{AB,\text{uNMA}}$. In this notation, uNMA is used to represent the updated NMA.

As A and B are already in the existing trial network, we can leverage indirect information from the existing network to reduce the standard error of the estimated effect size between A and B. Let the estimate of the effect size of B to A and its variance calculated from the existing network be $y_{AB,\text{eNMA}}$ and $\sigma^2_{AB,\text{eNMA}}$, respectively. In this notation, eNMA is used to represent the existing NMA. Let $y_{AB}$ be the estimated relative effect size and $\sigma^2_{AB,\text{trial}}$ be the variance of $y_{AB}$ when we analyze the new study in isolation, in other words, in the traditional way. Without loss of generality, suppose the baseline treatment is A. We denote the variance-covariance matrix of the updated trial network as $S^*$ and the updated design matrix as $X^*$.

$$S^* = \begin{bmatrix} \boldsymbol{S} & \boldsymbol{0} \\ \boldsymbol{0} & \sigma^2_{AB,\text{eNMA}} \end{bmatrix}, \quad X^* = \begin{bmatrix} \boldsymbol{X} \\ \boldsymbol{X}_{new} \end{bmatrix}.$$

where $\boldsymbol{X}_{new}$ is the design matrix of the new trial.

By Eq (3), the variance of $\hat{\mu}_{AB,\text{uNMA}}$ is given by

$$\sigma^2_{AB,\text{uNMA}} = \text{Var}(\hat{\mu}_{AB,\text{uNMA}}) = \left( \frac{1}{\sigma^2_{AB,\text{trial}}} + \frac{1}{\sigma^2_{AB,\text{eNMA}}} \right)^{-1}. \tag{4}$$

Obviously, we only need to minimize $\sigma^2_{AB,\text{trial}}$ in order to minimize $\text{Var}(\hat{\mu}_{AB,\text{uNMA}})$ given the fixed existing network. Therefore, even though the variances of the estimated effect size are

different when we analyze the new trial with or without the existing network, the optimal sample size for each method is the same. This characteristic is attributable to the optimization problem described in Eq (2).

## Power calculation for trial with pre-planned analysis with NMA

In this section we present the formula for how to estimate the power for a fixed total sample size when the intention is to analyse the trial result within an NMA i.e., the probability that the effect size on log odds ratio scale between two treatments will be statistically significant under a specific alternative hypothesis. Let the null hypothesis $H_0$ for the comparisons $AB$ be $\mu_{AB} = 0$. Let $\mu_{AB} \neq 0$ be the alternative hypothesis $H_1$. Define the significance level to be 0.05, then the expressions for the power for the alternative hypothesis is given by

$$\text{Power} = \Phi\left(\frac{lor}{s.d.} - z_{0.975}\right) + \Phi\left(-\frac{lor}{s.d.} - z_{0.975}\right), \tag{5}$$

where $lor$ denotes the log odds ratio between treatment A and B; $s.d$ denotes the population standard deviation of the effect size $lor$; $z_{0.975}$ is the 0.975th quantile of the normal distribution; $\Phi(.)$ denotes the cdf of the standard normal distribution.

Consistent with the notation used in previous sections, $\pi_i$ and $n_i$ denote the risk and the sample size of treatment i, $i \in A, B$, $\sigma^2_{AB, \text{eNMA}}$ represents the estimated variance of the relative effect for comparison $AB$. $lor$ is calculated based on $\pi_A$ and $\pi_B$. As $s.d.$, the standard deviation of the effect estimate, is unknown and we use the standard error of the estimated effect size ($\sigma_{AB,\text{uNMA}}$) of the A and B comparison from the NMA instead. The standard errors are different when we analyze the new trial with or without the existing network. Gathering the values of $\sigma^2_{AB,eNMA}$ from the existing network, the risk of each treatment, and the sample size of each treatment in the new trial, we can construct expressions for the estimates of $s.d$ by applying Eqs (1) and (4) as

$$\hat{s.d} = \begin{cases} \left[\dfrac{1}{n_A \pi_A (1-\pi_A)} + \dfrac{1}{n_B \pi_B (1-\pi_B)}\right]^{\frac{1}{2}} & \text{analysis without} \\[2mm] & \text{the existing network} \\[4mm] \left[\dfrac{n_A n_B \pi_A \pi_B (1-\pi_A)(1-\pi_B)}{n_A \pi_A (1-\pi_A) + n_B \pi_B (1-\pi_B)} + \dfrac{1}{\sigma^2_{AB,\text{eNMA}}}\right]^{-\frac{1}{2}} & \text{analysis with} \\[2mm] & \text{the existing network} \end{cases} \tag{6}$$

## Sample size calculation for trial with pre-planned analysis with NMA

Given the power formula in the previous section, it is straightforward to write the optimization problem of the scenario where a pre-specified power is the goal and the researchers intend to analyze the new trial with the existing network. Suppose the pre-specified power is 0.8, the optimization problem is

$$\underset{(n_A, n_B)}{\text{minimize}} \quad f(n_A, n_B) = n_A + n_B$$

$$\text{subject to} \quad \Phi\left(\frac{lor}{s.d} - z_{0.975}\right) + \Phi\left(-\frac{lor}{s.d} - z_{0.975}\right) \geq 0.8 \tag{7}$$

$$n_A \text{ and } n_B \text{ are positive integers.}$$

where

$$s.d = \left[ \frac{n_A n_B \pi_A \pi_B (1 - \pi_A)(1 - \pi_B)}{n_A \pi_A (1 - \pi_A) + n_B \pi_B (1 - \pi_B)} + \frac{1}{\sigma^2_{AB,\text{eNMA}}} \right]^{-\frac{1}{2}} .$$

## Application and simulation

Here we conduct two simulation studies to assess the proposed method to design a new two-arm trial while borrowing information from the existing network and comparing it to the traditional methods. In Dataset description, we describe the existing NMA dataset of Bovine Respiratory Disease (BRD) in our application. Simulation I aims to evaluate analyzing the new trial with and without the existing network given a fixed total sample size. We also include an evaluation of the allocation of study participants to treatment group i.e. optimal versus even to determine if this has an impact on the power of the new trial. Through the value of power, standard error (SE) of the estimated effect size, we evaluate differences in power between analysis with and without the network and optimal allocation and even allocation. Simulation II generates the optimal sample size allocation given the condition that the power is at least 80% when analyzing the new trial with the existing network, and compares this sample size required to the traditional design method without the existing network. Here we also look into two optimal sample size strategies for the binary outcome, one is the optimal sample size without the constraint of even allocation, the other is the optimal sample size with the constraint of even allocation. The required total sample size, value of power, and SE of the estimated effect size are calculated to compare different sample size allocation strategies and different analysis methods.

## Dataset description

A previously published network of interventions for the treatment of Bovine Respiratory Disease (BRD) in feedlot cattle is used for the simulation study [14]. The dataset is comprised of 98 trials, 13 treatments, and 204 arms. Most trials contain two arms and eight trials contain three arms. A network plot is shown in Fig 3. Arm-level data are available and the outcome is a dichotomous measurement. To compare treatments, the log odds ratios between pairwise comparisons are calculated.

## Simulation I

This section describes how a new trial would be simulated with the optimal or even allocation, and evaluates the performance of analyzing the trial with or without existing network.

The new two-arm trial contains Ceftiofur crystalline acid administered in the pinna (CEFTP) and Tildipirosin administered subcutaneous (TILD), both at the manufacturers recommended dose and regime. These two treatments already appear in the existing trial network but do not have a direct comparison. We set two effect sizes (0.299, 0.5) between CEFTP and TILD on the log odds ratio scale and five total sample sizes (100, 200, 500, 1000, 2000) to formulate 10 scenarios. To simplify the notation, we replace CEFTP and TILD with A and B. For each scenario with total sample size *n* and effect size *lor*, the simulation process is as follows:

1. From a network meta-analysis of the existing network, the risk of treatment *j* is estimated and denoted as $p_j$. The risk of B is re-calculated according to the risk of A and the effect size *lor* set.

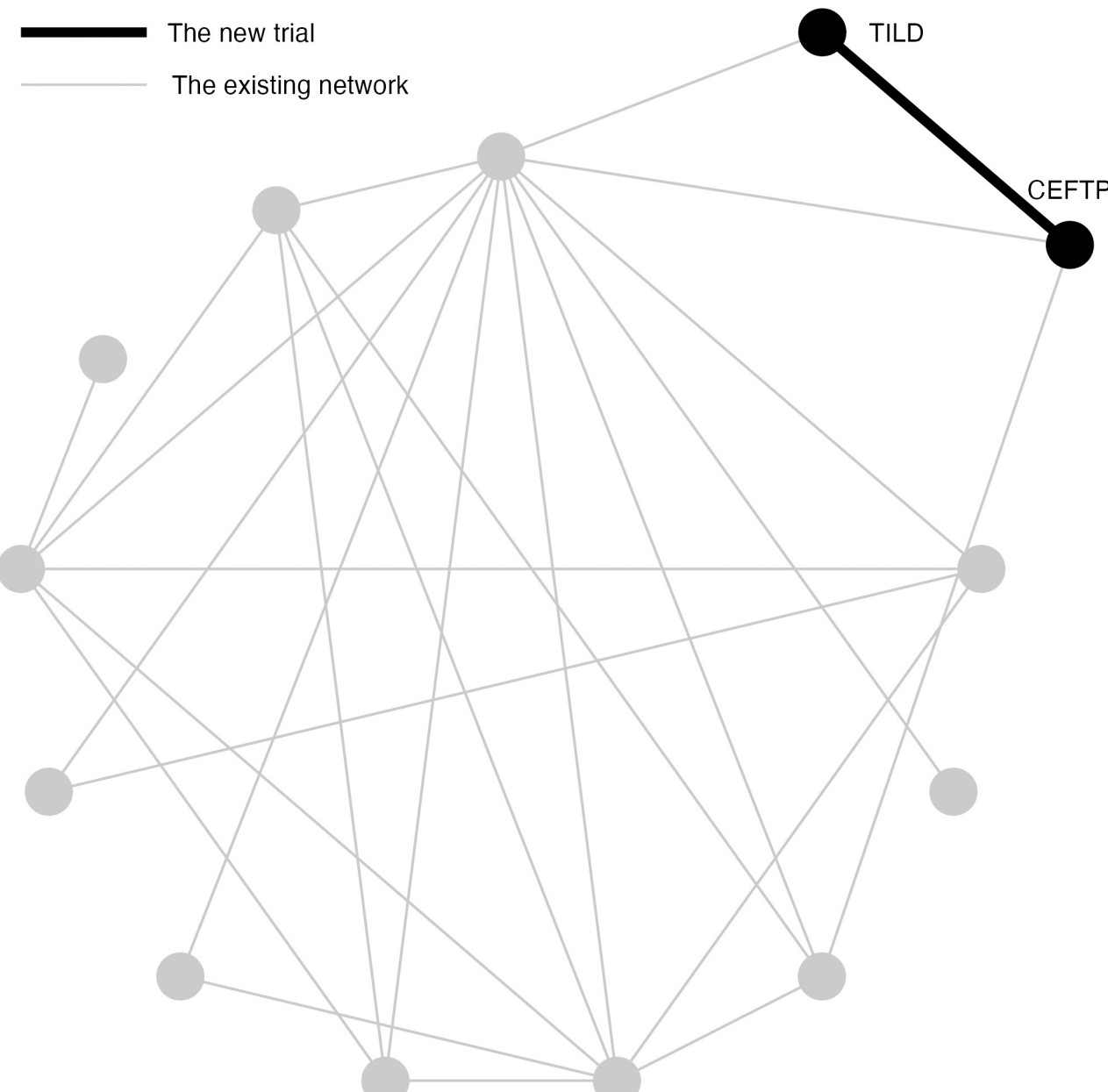

**Fig 3. An existing network of interventions for the treatment of Bovine Respiratory Disease (BRD) in feedlot cattle with a new study between CEFTP(A) and TILD(B).** Each node represents a treatment, with a total of 13 treatments in the existing network. A line connecting two treatments indicates a direct comparison between them. The grey lines represent trials in the existing network, while the black line represents the new trial connecting CEFTP and TILD.

2. For each treatment group with treatment $j$ and total sample size $n_{s,j}$ in study $s$ of the existing data, replace the number of events $r_{s,j}$ with a random number generated from Binom($n_i,p_j$). This is the simulated existing data.

3. Let $p_A$ and $p_B$ denote the risks of A and B calculated in Step 1. We find the optimal animal allocation ($n_A$, $n_B$) by solving the optimization problems in Eq (2).

4. Data representing the new study is generated by sampling $r_i$ from Binom($n_i$, $p_i$), $i \in \{A, B\}$.

5. The simulated data are analyzed by logistic regression. We use the term "analyze without existing network" in the following contents to denote this step.

6. The point estimate and its standard error for the comparison of A and B are extracted from the result.

7. Use a 0-1 indicator to denote if there is a significant difference between the comparison of A and B ($\alpha = 0.05$).

8. We then calculate the power and standard error of the estimated relative effect size using the formula in Eqs (5) and (6)

9. The point estimate and its standard error for the comparison of A and B from step 5 are added to the simulated existing data to represent a row of study-level data. We use network meta-analysis to analyze the combined data, which refers to "analyzed with existing network" in the following contents. Repeat Step 6 to 8 for the new result, which is generated by analyzing with existing network.

10. We then re-generate the new study by sampling $r_i$ from Binom($n/2$, $p_i$), $i \in \{A, B\}$. Repeat Step 5 to 9 for this new study.

11. Repeat steps 2-10 10,000 times. Calculate the proportion of the indicator equal to 1 to obtain the simulation power and average the powers in Step 8 to obtain the formula power. Likewise, obtain the simulation standard error and the formula standard error.

## Simulation II

We explain the process of simulation to evaluate the performance of analyzing the new trial with or without existing network when the optimal allocation for each treatment is determined the mimimum total sample size required to reach the pre-specified power.

The two treatments in the new trial are the same as Simulation I and the desired power is 80%. We set four effect sizes (0.299, 0.4, 0.5, 0.6) for four simulation scenarios. For each scenario with the effect size *lor*, the process is conducted as below:

1. From a network meta-analysis of the existing network, the risk of treatment $j$ is estimated and denoted as $p_j$. The risk of B is re-calculated based on the risk of A and the effect size *lor* set.

2. For each treatment group with treatment $j$ and total sample size $n_{s,j}$ in study $s$ of the existing data, replace the number of events $r_{s,j}$ with a random number generated from Binom($n_i, p_j$). This is the simulated existing data.

3. Consider analyzing the new trial with the existing network, find the optimal animal allocation ($n_A$, $n_B$) by solving the optimization problems in Eq (7).

4. Data representing the new study is generated by sampling $r_i$ from Binom($n_i$, $p_i$), $i \in \{A, B\}$.

5. Analyze the simulated new trial using logistics regression.

6. The point estimate and its standard error are extracted from the result.

7. Use an 0-1 indicator to denote if there is a significant difference between the comparison ($\alpha = 0.05$).

8. The result from step 5 is added to the simulated existing data to represent a row of study-level data. We use network meta-analysis to analyze the combined data. Repeat step 6 and 7 for the new result generated by analyzing with existing network.

9. Re-generate the sample size for each treatment by add the constraint of even allocation to the optimization problem in Eq (7). Repeat step 4-8 for this new study.

10. Repeat steps 2-9 for 10,000 times. Calculate the proportion of the indicator equal to 1 to get the simulation power. Likewise, we average the sample size $n_A$, $n_B$, and standard error across 10,000 times.

## Results

This section summarizes the results of the two simulation studies described previously. The results of simulation I are shown in Table 1. For each fixed total sample size and log odds ratio combination, the results are represented when the new trial has been analyzed with and without the existing network and for two sample size allocation strategies: even allocation and the optimal allocation. We report the following four metrics: simulation power, formula power, simulation SE, formula SE. Power and SE are calculated using two methods: directly from the formula, and, summarized from simulation, to verify the correctness of our simulation process. The simulation power and simulation SE should be equal to the formula power and formula SE as long as the iteration of the simulation is large enough.

In Table 1 we can see the large difference in power between the two analysis methods. As expected, under the same sample size allocation and the log odds ratio, analyzing the new trial with the existing network is more powerful than without. This increase in power is directly attributed to the smaller SE of analyzing with the existing network. As is shown in Table 1, the differences in power, and standard error between the two sample size allocation strategies are minimal. The magnitude of the difference between two allocation strategies in the power or SE is too small to detect due to the randomness of the simulation process. For example, when the total sample size is 500 and the log odds ratio is 0.299, the simulation power is 46.41% for the even allocation while it is 46.13% for the optimal allocation when we analyze the new trial with the existing network.

The results of simulation II are shown in Table 2. The risk of CEFTP is a constant value, which is estimated from the existing network. For each log odds ratio between CEFTP and TILD, again the results represent analysis with and with the network and the two sample size allocation strategies. For both allocation strategies, our goal is to achieve at least 80% power under the condition of analyzing with the existing network. For each analysis, we calculate the following two metrics: power and SE. We can see that as before leveraging the network increases power. When analyzing the new trial with the existing network the power is about 80% because it is our goal when calculating the optimal sample size. The power of analyzing without the existing network is about 70% and 42% for the log odds ratio 0.299 and 0.5, respectively. This is expected from the result of Simulation I. Under the same sample size for each treatment, the SE of analyzing without the existing network is always larger than that of with, so that the power of analyzing without the existing network is always smaller than that of with. Another aim of Simulation II is to compare the optimal allocation and even allocation. Under our simulation setting, we did not observe any obvious difference in the total sample sizes needed. For example, when the log odds ratio is 0.299, the average total sample size needed across 10,000 simulations is 1666 when no restriction is added to the optimization problem, while it is 1668 when the even allocation requirement is added.

**Table 1. Comparison of power and standard error for the new trial under different analysis methods and different sample size allocation strategies given the total sample size.** Simulation results are based on 10000 samples.

| sample size | log odds ratio | allocation | with existing network | | | | without existing network | | | |
|---|---|---|---|---|---|---|---|---|---|---|
| | | | power | | SE | | power | | SE | |
| | | | simulation[1] | formula[2] | simulation[3] | formula[4] | simulation | formula | simulation | formula |
| 100 | 0.299 * | (50,50) | 28.77% | 29.89% | 0.2098 | 0.2089 | 8.37% | 9.33% | 0.5085 | 0.4916 |
| 100 | 0.299 | (52,48) | 28.82% | 29.89% | 0.2098 | 0.2089 | 8.72% | 9.34% | 0.5077 | 0.4912 |
| 100 | 0.5 | (50,50) | 67.44% | 68.30% | 0.2061 | 0.2053 | 17.29% | 18.00% | 0.4969 | 0.4810 |
| 100 | 0.5 | (53,47) | 67.59% | 68.34% | 0.2060 | 0.2052 | 17.39% | 18.06% | 0.4951 | 0.4800 |
| 200 | 0.299 | (100,100) | 33.34% | 34.30% | 0.1930 | 0.1923 | 13.69% | 13.81% | 0.3532 | 0.3476 |
| 200 | 0.299 | (104,96) | 33.48% | 34.32% | 0.1930 | 0.1922 | 13.79% | 13.83% | 0.3528 | 0.3473 |
| 200 | 0.5 | (100,100) | 75.30% | 75.45% | 0.1895 | 0.1888 | 32.16% | 31.24% | 0.3453 | 0.3401 |
| 200 | 0.5 | (107,93) | 75.23% | 75.50% | 0.1893 | 0.1887 | 31.90% | 31.35% | 0.3443 | 0.3394 |
| 500 | 0.299 | (250,250) | 46.41% | 46.75% | 0.1596 | 0.1592 | 27.28% | 27.47% | 0.2211 | 0.2199 |
| 500 | 0.299 | (261,239) | 46.13% | 46.79% | 0.1595 | 0.1591 | 26.99% | 27.51% | 0.2209 | 0.2197 |
| 500 | 0.5 | (250,250) | 89.05% | 89.30% | 0.1565 | 0.1561 | 64.94% | 64.22% | 0.2164 | 0.2151 |
| 500 | 0.5 | (267,233) | 89.64% | 89.36% | 0.1564 | 0.1559 | 65.13% | 64.41% | 0.2159 | 0.2146 |
| 1000 | 0.299 | (500,500) | 63.57% | 64.02% | 0.1291 | 0.1289 | 48.22% | 48.54% | 0.1558 | 0.1555 |
| 1000 | 0.299 | (522,478) | 63.52% | 64.07% | 0.1291 | 0.1289 | 47.21% | 48.61% | 0.1557 | 0.1553 |
| 1000 | 0.5 | (500,500) | 97.58% | 97.71% | 0.1266 | 0.1264 | 90.77% | 90.77% | 0.1525 | 0.1521 |
| 1000 | 0.5 | (534,466) | 97.61% | 97.74% | 0.1263 | 0.1262 | 91.14% | 90.90% | 0.1521 | 0.1518 |
| 2000 | 0.299 | (1000,1000) | 85.24% | 85.38% | 0.0993 | 0.0992 | 77.85% | 77.63% | 0.1101 | 0.1099 |
| 2000 | 0.299 | (1043,957) | 85.30% | 85.43% | 0.0992 | 0.0992 | 78.12% | 77.71% | 0.1099 | 0.1098 |
| 2000 | 0.5 | (1000,1000) | 99.94% | 99.93% | 0.0973 | 0.0972 | 99.68% | 99.64% | 0.1077 | 0.1076 |
| 2000 | 0.5 | (1067,933) | 99.94% | 99.93% | 0.0971 | 0.0970 | 99.70% | 99.65% | 0.1074 | 0.1073 |

\* We set 0.299 as one of the log odds ratios in simulation scenarios because it is the estimated log odds ratio between CEFTP and TILD from the existing network.

[1] The simulation power is calculated directly from the simulation by the proportion of the significance indicator equal to 1.

[2] The formula power is the average power across 10,000 formula power calculations. In each simulation round, power was calculated based on Eq (5) and then averaged across 10,000 rounds to obtain the formula power.

[3] The simulation SE is the SE averaged across 10,000 SEs, with each standard error calculated based on the actual simulated data from the corresponding simulation round.

[4] The formula SE is the SE averaged across 10,000 formula SEs, with each standard error calculated based on Eq (6) using the true parameter values.

**Table 2. Comparison of required total sample size, power, and standard error for the new trial under different analysis methods and different sample size allocation strategies given the power is at least 80% when analyze with existing network.** Simulation results are based on 10000 samples.

| lor | Risk | | Sample Size | | | with existing network | | without existing network | |
|---|---|---|---|---|---|---|---|---|---|
| | CEFTP | TILD | CEFTP | TILD | Total | power | SE | power | SE |
| 0.299 * | 0.1876 | 0.2374 | 834 | 834 | 1668 | 80.2% | 0.107 | 69.7% | 0.121 |
| 0.299 | 0.1876 | 0.2374 | 869 | 797 | 1666 | 80.4% | 0.107 | 70.5% | 0.120 |
| 0.4 | 0.1876 | 0.2562 | 354 | 354 | 708 | 80.0% | 0.143 | 59.2% | 0.183 |
| 0.4 | 0.1876 | 0.2562 | 373 | 333 | 706 | 80.5% | 0.143 | 59.3% | 0.183 |
| 0.5 | 0.1876 | 0.2757 | 139 | 139 | 278 | 80.2% | 0.179 | 41.5% | 0.292 |
| 0.5 | 0.1876 | 0.2757 | 148 | 129 | 277 | 80.0% | 0.179 | 41.7% | 0.292 |
| 0.6 | 0.1876 | 0.2961 | 24 | 24 | 48 | 80.1% | 0.215 | 11.9% | 0.752 |
| 0.6 | 0.1876 | 0.2961 | 26 | 22 | 48 | 80.3% | 0.215 | 12.0% | 0.750 |

\* We set 0.299 as one of the log odds ratios in simulation scenarios because it is the estimated log odds ratio between CEFTP and TILD from the existing network.

## Application tool: R Shiny webpage

Our goal here is the introduce two tools that can help researchers readily use the approaches we have presented. We do this with an application and an R package. The application can be accessed from https://fangshu.shinyapps.io/CalSampleSize/. The screenshot of two examples are shown in Figs 4 and 5. We will provide details of the steps and data required to use the R Shiny web page below.

### Step 1

In step 1, a dataset representing the existing evidence should be uploaded. A sample dataset is provided and the uploaded dataset should have the same format as the sample dataset. Note here we use the contrast-level data.

### Step 2

Step 2 is for choosing the treatments in your future two-arm trial. The options wouldn't appear unless you complete step 1, and it contains all the treatments appeared in the existing evidence. The order of choosing the treatment doesn't matter. When you select the second treatment, it will automatically eliminate the one you choose first so that you don't bother with choosing the same two treatments.

### Step 3

Step 3 is to assign the values of parameters used for calculating the optimal sample size. Since we use the contrast-level data, it's impossible to get the risk of each treatment unless we know

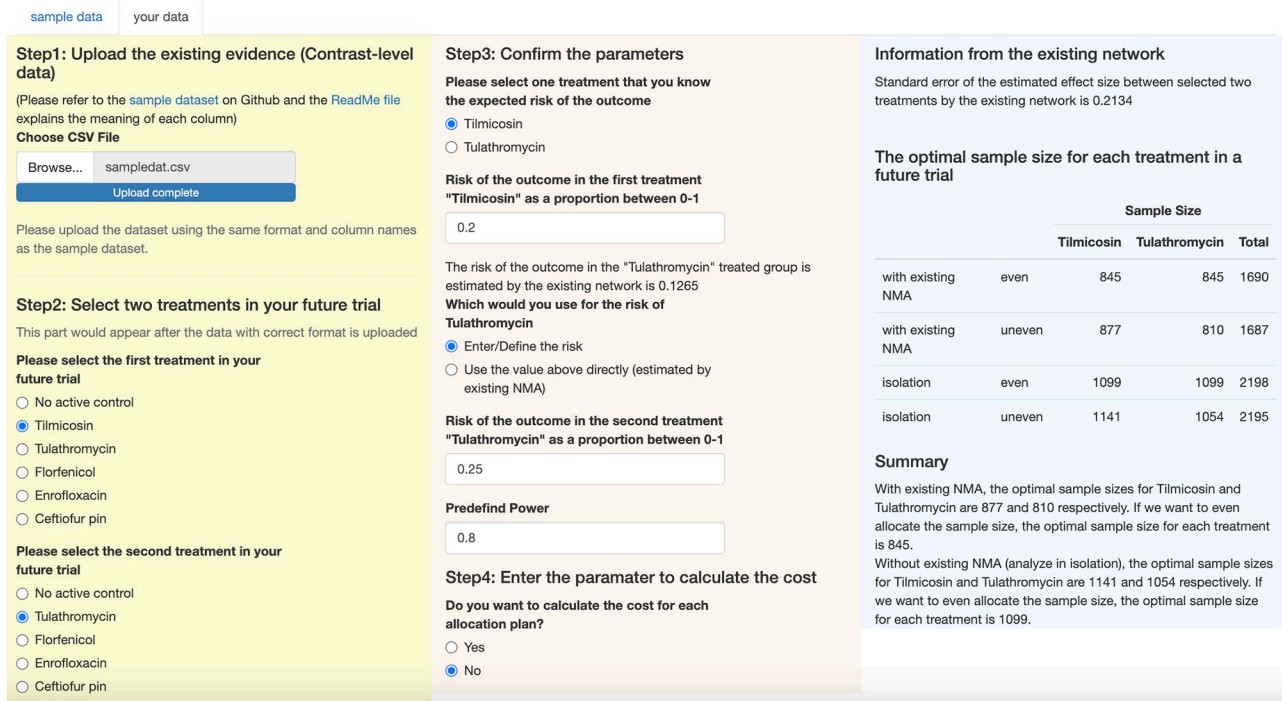

**Fig 4. An example of using the R Shiny app webpage without the cost output selected.** A sample dataset is provided for ease of use.

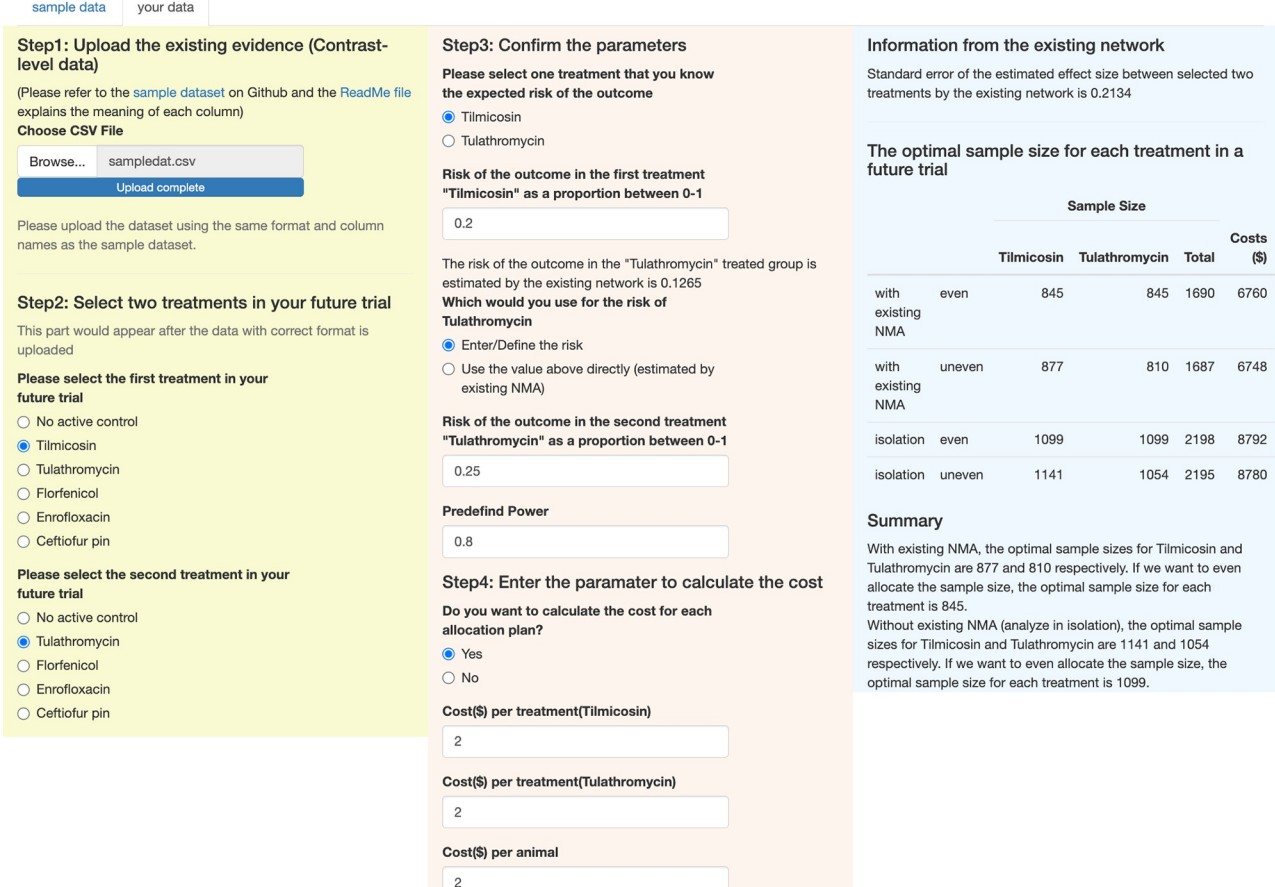

**Fig 5. An example of using the R Shiny app webpage with the cost output.** A sample dataset is provided for ease of use.

the risk of one treatment. Therefore, you need to input the risk of one treatment. After entering the risk, it will tell you the risk of the other treatment combining the information of existing network and your input risk. It's up to you to use the risk of the other treatment from the existing network or enter the risk by yourself. Now we have the risk of two treatments in hand, all we need to calculate the optimal sample size is a pre-specified power. The default value is 0.8 but you can change it to fit your need.

### Step 4

Step 4 is irrelevant to the optimal sample size calculation, and it's for showing the cost of different strategies (with/without existing NMA and even/uneven allocation). If the cost of each animal or each treatment is not clear at this moment, you can choose 'No' to avoid further input. Otherwise, you need to input the cost per treatment and per animal in this step.

### Output panel

The output panel consists of three parts. At the top, the standard error of estimated log odds ratio between two treatments by existing network is given. In the middle, the optimal sample

size for each treatment is calculated under four scenarios: analyze with existing network; analyze with the existing network and restrict the allocation to be the same for two treatments; analyze in isolation; analyze in isolation and restrict the allocation to be the same for two treatments. Note that the table has an additional column of costs if the costs are entered in step 4. At the bottom, there is a brief summary of the table above.

## The package

The methods outlined in the previous sections are implemented in an R package called OssaNMA. The package can be used for using information from existing network meta-analyses to optimize or allocate the sample size of a new two-arm trial with a binary outcome. Two functions and one example dataset are included in the package.

### ssnma

The first main function is the following:

```
ssnma(p1, p2, enma_sigma, power.level, sig.level = 0.05,
  method = "with", allocation = "uneven")
```

The available arguments are:

- `p1` Risk of treatment 1

- `p2` Risk of treatment 2

- `enma_sigma` Standard error of the estimated effect size (log odds ratio) between treatment 1 and treatment 2 from the existing network

- `power.level` Power of test we want to obtain

- `sig.level` Significance level, the default value is 0.05

- `method` a character string specifying the method of analyzing the new trial, must be one of 'with' (default) or 'without'

- `allocation` a character string specifying the type of sample size allocation between two groups, must be one of 'uneven' (default) or 'even'

This function returns a list with the following components:

- `sample_size` Sample size for each treatment group

- `power` Power of the test

Assuming a new two-arm trial comparing treatment 1 and treatment 2 is to be planned. The two treatments exist in the existing network, which serves as a foundation to analyze the new trial with the existing network using NMA.

Given that the risk of treatment 1 is 0.2, the risk of treatment 2 is 0.3, and the standard error of the estimated effect size between two treatments from the existing NMA is 0.3, `ssnma()` can be applied to solve the minimum required total sample size for the new trial to achieve a power of 0.8 and allocate it to each treatment group under different allocation method (even or uneven) and analysis method (with or without the existing network).

The optimal sample size for each treatment group when the new trial is analyzed with the existing network is the following:

```
R> ssnma(p1 = 0.2, p2 = 0.3, enma_sigma = 0.3, power.level = 0.8,
+ sig.level = 0.05, method = "with", allocation = "uneven")

$sample_size
[1] 187 163

$power
[1] 0.801
```

The optimal sample size under even allocation for each treatment group when the new trial is analyzed with the existing network is the following:

```
R> ssnma(p1 = 0.2, p2 = 0.3, enma_sigma = 0.3, power.level = 0.8,
+ sig.level = 0.05, method = "with", allocation = "even")

$sample_size
[1] 176 176

$power
[1] 0.801
```

The optimal sample size under even allocation for each treatment group when the new trial is analyzed without the existing network is the following:

```
R> ssnma(p1 = 0.2, p2 = 0.3, enma_sigma = 0.3, power.level = 0.8,
+ sig.level = 0.05, method = "without", allocation = "uneven")

$sample_size
[1] 317 277

$power
[1] 0.801

R> ssnma(p1 = 0.2, p2 = 0.3, enma_sigma = 0.3, power.level = 0.8,
+ sig.level = 0.05, method = "without", allocation = "even")

$sample_size
[1] 298 298

$power
[1] 0.801
```

### ssanma

The second main function is the following:

```
ssnma(p1, p2, enma_sigma, power.level, sig.level = 0.05,
  method = "with", allocation = "uneven")
```

The available arguments are:

- p1 Risk of treatment 1

- p2 Risk of treatment 2

- `enma_sigma` Standard error of the estimated effect size (log odds ratio) between treatment 1 and treatment 2 from the existing network

- `N`: Number of total sample size

- `sig.level` Significance level, the default value is 0.05

- `method` a character string specifying the method of analyzing the new trial, must be one of 'with' (default) or 'without'

- `allocation` a character string specifying the type of sample size allocation between two groups, must be one of 'uneven' (default) or 'even'

   This function returns a list with the following components:

- `sample_alloc` Sample size allocation to each treatment group

- `power` Power of the test

   Assume that we have the same new trial planned as section ssnma, the goal in this section is to calculate the optimal sample size allocation to each treatment group with a fixed total sample size of 200 to maximize the power, `ssanma()` is used and it covers the allocation under different allocation method (even or uneven) and analysis method (with or without the existing network). Take the uneven allocation for example, the optimal allocations under both analysis methods are the following:

```
R> ssanma(p1 = 0.2, p2 = 0.3, enma_sigma = 0.3, N = 200, sig.
level = 0.05,
+ method = "with")

$sample_alloc
[1] 107 93

$power
[1] 0.679

R> ssanma(p1 = 0.2, p2 = 0.3, enma_sigma = 0.3, N = 200, sig.
level = 0.05,
+ method = "without")

$sample_alloc
[1] 107 93

$power
[1] 0.37
```

   The optimal sample allocation when we analyze the new trial traditionally is the same as analyzing it with the existing network, as explained in Materials and methods. However, the power decreased greatly compared to analyzing it with the existing network.

## Empirical illustrations

We consider the problem of working with the identical dataset as introduced in Dataset description. After loading the OssaNMA package and additional netmeta [15] package to

calculate the standard error of the estimated effect size, we load the example data from the OssaNMA as the following:

```
R> data("BRDdat")
R> head(BRDdat)
#>   studlab            treat1       treat2       TE      seTE
#> 1       1 No active control  Florfenicol 1.817766 0.4198201
#> 2       2 No active control Enrofloxacin 3.471966 1.5027924
#> 3       3 No active control Enrofloxacin 3.201584 0.4781349
#> 4       4 No active control Enrofloxacin 1.434531 0.3878773
#> 5       5 No active control Enrofloxacin 3.197472 0.4938823
#> 6       6 No active control Enrofloxacin 1.945910 0.3854496
```

In the dataset, each row represents the summary statistics for a pairwise comparison between two treatments in a trial. See the meaning of each column below:

- studlab: study id

- treat1: name of treatment 1

- treat2: name of treatment 2

- TE: estimated treatment effect size (log odds ratio) between treat1 and treat2

- seTE: standard error of TE

Let's conduct a network meta-analysis using this dataset:

```
R> library("netmeta")
R> nma_res <- netmeta(TE, seTE, treat1, treat2, studlab,
+ data=BRDdat, sm="OR", comb.fixed = T, comb.random = F)
```

Assuming a new two-arm trial comparing CEFTP and TILD is to be planned. To apply the functions in OssaNMA to help to plan the new trial, we need to have the standard error of the estimated effect size between the two treatments, CEFTP and TIILD, from the existing network. We can get the value by:

```
R> enma_sigma <- nma_res$seTE.fixed['Ceftiofur pin','Tildipirosin']
R> enma_sigma
```

Also, we need to know the risk of two treatments in the new trial. Some options are:

- Taking No active control (NAC) as a baseline treatment, we can calculate the estimated log odds ratio between NAC and other treatments using NMA. As for the risk of NAC, we can get it by pooling the arm-level data from the existing network if any.

- Use other sources of evidence to assign the risk of two treatments.

Take the first option for example:

```
R> # The risk of NMA is calculated by pooling the arm-level data from
R> # the existing network. The arm-level data is not provided in the
package
R> # so the value is given directly here.
R> p_nac <- 0.68
R> # extract the log odds ratio between NAC and two treatments from
nma_res
```

```
R> lor_nac_enro <- nma_res$TE.fixed['No active control','Ceftiofur pin']
R> lor_nac_flor <- nma_res$TE.fixed['No active control','Tildipirosin']
R> # calculate risk of Ceftiofur pin, name it as p1
R> p1 <- p_nac/(p_nac + exp(lor_nac_enro)*(1-p_nac))
R> # calculate risk of Tildipirosin, name it as p2
R> p2 <- p_nac/(p_nac + exp(lor_nac_flor)*(1-p_nac))
```

This is the value of p1 and p2:
```
R> p1
[1] 0.1868955
R> p2
[1] 0.2366667
```

With p1, p2, and enma_sigma obtained from the existing NMA, we can solve the minimum required total sample size for the new trial to achieve a pre-specified power using ssnma() or calculate the optimal sample size allocation to each treatment group with a fixed total sample size to maximize the power using ssanma(), as how we applied the two functions in the previous section when we have specified values of the input parameters.

See the application using the p1, p2, and enma_sigma obtained from the existing NMA below:

The first example shows that to solve the minimum required total sample size for the new trial to achieve a pre-specified power of 0.8 and allocate it to each treatment group when the new trial is analyzed with the existing network, ssnma() can be applied as the following:
```
R> ssnma(p1 = p1, p2 = p2, enma_sigma = enma_sigma, power.level = 0.8,
+ sig.level = 0.05, method = "with", allocation = "uneven")

$sample_size
[1] 853 782

$power
[1] 0.8
```

The second example shows that to calculate the optimal sample size allocation to each treatment group with a fixed total sample size of 800 to maximize the power when the new trial is analyzed with the existing network, ssanma() is used as the following:
```
R> ssanma(p1 = p1, p2 = p2, enma_sigma = enma_sigma, N = 800,
+ sig.level = 0.05, method = "with")

$sample_alloc
[1] 417 383

$power
[1] 0.588
```

## Discussion

Network meta-analysis of the existing evidence provides a quantitative framework to enlighten the design of new trials. Several previous studies have discussed approaches to planning future

studies based on the network meta-analysis [2, 4, 5]. These papers focused on designing a series of trials. While in this paper, the focus is on a two-arm trial where both treatments have been included in the existing network but there was no direct comparison. Clearly, designing one trial is more realistic than designing a series of trials. Additionally, this paper focuses on the new trial itself instead of the result of the updated meta-analysis. In this paper, we purposed a method to calculate the optimize sample allocation to improve the power of the comparison of interest in the new trial given the sample size fixed. As expected, leveraging information from existing network can improve power or reduce required sample size. Besides, we found that the difference between even allocation and optimal allocation is little. Since we focused on a specific future trial, the formulas described in this paper are easy to be implemented. To make our method more feasible to trial planners, an R Shiny app and OssaNMA package are deployed to calculate the optimal sample size.

In planning future trials, under the condition that a well-conducted NMA is available for leveraging information, our methods have primary use in determining the sample size needed for each treatment when our goal is to achieve a pre-specified power. It could also be used in determining the sample size allocation when our goal is to maximize the power when the total sample size is fixed. Though we showed that the even and uneven allocation have no obvious difference in power, it is still worth exploring in application if the cost of each treatment is taken into consideration.

There are some limitations in this paper. First, the goal of this paper is to improve the power of a new trial when the total sample size is fixed or to reduce the sample size needed when a power level is pre-specified to achieve, but there are other aspects we didn't consider such as the precision of estimates, the costs of different treatments. Second, as the methods described in this paper are based on network meta-analysis, it is only applicable to cases where a well-conducted NMA is available and there is no big variability across trial settings. Third, we only focus on a specific trial design in this paper, which is, two-arm trial with both treatments in the existing network but didn't have direct comparison. Though it is quite common, there are still other kinds of designs that are worth exploring. For example, a new trial involved a new treatment.

Future research related to this paper could be conducted in the following directions. First, instead of considering a new trial that contains two existing treatments, it would be useful to understand how to design a new trial that contains one existing treatment and one new treatment. Second, leveraging information from existing network to inform the design of new trial could break the assumed lack of dependence between studies. In particular, if a study is performed because of a statistically significant finding in an early study, this may lead to some potential problems. Further investigation needs to be conducted.

## Conclusion

To conclude, first, borrowing information from the existing network meta-analysis is resource-saving compared with analyzing without it when our goal is to achieve a pre-specified power. Secondly, borrowing information from the existing network meta-analysis increases power compared with analyzing without the existing network when the total sample size is fixed. Thirdly, the optimal sample size allocation for each treatment is uneven for binary outcomes; however, this difference between the even and uneven allocation is practically ignorable. The last statement is true for both analysis methods (with/without the existing network) and both goals (the optimal allocation for fixed sample size or a pre-specified power). Lastly, This paper has presented the R package OssaNMA which provides a convenient set of tools for calculating the minimum total sample size needed to achieve a pre-specified power or the

optimal allocation for each treatment group with a fixed total sample size to maximize the power. Examples illustrating the use of the package in practical applications have been presented. Besides, our R Shiny app provides a means to implement the above calculations and is freely available to practitioners. We encourage clinical trial planners to utilize our package or the online tool for optimal sample size/allocation calculation to achieve more efficient and powerful trial designs.

## Author Contributions

**Conceptualization:** Chong Wang, Annette M. O'Connor.

**Data curation:** Fangshu Ye.

**Formal analysis:** Fangshu Ye.

**Investigation:** Fangshu Ye, Chong Wang, Annette M. O'Connor.

**Methodology:** Fangshu Ye, Chong Wang, Annette M. O'Connor.

**Project administration:** Chong Wang, Annette M. O'Connor.

**Resources:** Annette M. O'Connor.

**Software:** Fangshu Ye.

**Supervision:** Chong Wang, Annette M. O'Connor.

**Validation:** Fangshu Ye.

**Visualization:** Fangshu Ye.

**Writing – original draft:** Fangshu Ye.

**Writing – review & editing:** Fangshu Ye, Chong Wang, Annette M. O'Connor.

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
