## [Decision Letter · Decision Letter 0]

14 Sep 2023

PONE-D-23-20293OssaNMA: An R Package for using information from network meta-analyses to optimize the power and sample allocation of a new two-arm trialPLOS ONE

Dear Dr. Wang,

Thank you for submitting your manuscript to PLOS ONE. After careful consideration, we feel that it has merit but does not fully meet PLOS ONE’s publication criteria as it currently stands. Therefore, we invite you to submit a revised version of the manuscript that addresses the points raised during the review process.

The work is solid overall. Please make any edits suggested by Reviewer 1 and submit a revised version as soon as possible. ==============================

We look forward to receiving your revised manuscript.

Kind regards,

Zechen Chong

Academic Editor

PLOS ONE

2. We note that Figures 4 and 5 in your submission contain copyrighted images. All PLOS content is published under the Creative Commons Attribution License (CC BY 4.0), which means that the manuscript, images, and Supporting Information files will be freely available online, and any third party is permitted to access, download, copy, distribute, and use these materials in any way, even commercially, with proper attribution. For more information, see our copyright guidelines: http://journals.plos.org/plosone/s/licenses-and-copyright.

1. You may seek permission from the original copyright holder of Figures 4 and 5 to publish the content specifically under the CC BY 4.0 license.

Reviewers' comments:

Reviewer's Responses to Questions

**Comments to the Author**

1. Is the manuscript technically sound, and do the data support the conclusions?

Reviewer #1: Yes

Reviewer #2: Yes

2. Has the statistical analysis been performed appropriately and rigorously? 

Reviewer #1: Yes

Reviewer #2: Yes

3. Have the authors made all data underlying the findings in their manuscript fully available?

Reviewer #1: No

Reviewer #2: Yes

4. Is the manuscript presented in an intelligible fashion and written in standard English?

Reviewer #1: Yes

Reviewer #2: Yes

5. Review Comments to the Author

Reviewer #1: Comments to OssaNMA: An R Package for using information from network meta-analyses to optimize the power and sample allocation of a new tow-arm trial.

The authors propose one new approach to maximizing power proposed by previous researchers to leverage prior evidence using meta-analysis (NMA) to inform the sample size determination of a new trial. They develop formulas to address these options and use simulations to validate their formula and evaluate the performance of different analysis methods in terms of power. They also implement the proposed method into the R package OssaNMA and publish an R Shiny app for the convenience of the application. The goal of the package is to enable researchers to readily adopt the proposed approach which can improve the power of an RCT and is therefore resource-saving. The method holds relevance and practical utility, particularly in the context of designing two-arm Randomized Controlled Trials (RCTs) for research purposes. The method and formulas seem work pretty well, but I have the following concerns:

1. This method is only tried on simulation analysis to compare the statistical power and standard error with existing network and without existing network. Have the authors ever applied this method on real clinical trial data by any chance to evaluate the power and standard error? I am considering simulation would be overfitting in some circumstances.

2. How did the authors select the existing network? Selecting an accurate and representative network from the existing options can be challengeable, given the potential for inaccuracies in available resources. Were there any specific protocols employed to navigate the incorporation of existing networks into this process?

3. How many existing network studies does it need at minimum to ensure a robust power of the new method? Will the statistical power of existing network meta-analysis affect the statistical power of the two-arm trial and sample size optimization?

4. Providing legends to elucidate your figures, such as in Figure 3, by explaining the significance of nodes and edges, would enhance clarity and comprehension.

5. The clarity of Figure 4 and Figure 5 is compromised due to their low resolution. The figures don’t not show clearly to me. It would be better if you can improve it with higher resolution.

Minor issues: (Confidential below: only for editor to see)

1. The problem is less important in the field. The method which focuses on statistical power in the research design is important but not a creative topic. The paper introduces method using existing network, which has advantages and disadvantages. Through simulation analysis, we can see dramatically increase of the power and less standard error in the existing network group compared to the group without existing network, but sometimes existing network can be biased, so this method can be a good way to consider as reference, not decisive to set the sample size.

2. Power is influenced by multiple interrelated factors, each of which can exert an impact. Moreover, diverse trials encompass varying circumstances, making real-world scenarios inherently complex. Furthermore, this paper does not establish the method's applicability and effectiveness in real clinical trials, leaving this aspect uncertain.

3. The package function seems not very functional and limited to two-arm clinical trials based on certain number of previous studies. Specially for some rare diseases or creative work, this method might not fit.

Reviewer #2: The study developed an R package and related Shiny app called OssaNMA for power and sample size optimization for a new two-arm trial. The tool was well-developed, and the Shiny app was easy to use, this could help researchers for future RCTs planning to get an optimal power and sample size. The findings and analysis presented in this study are beneficial to the research community.

Revision and suggestions:

1. The github page can have more example code shown in README.md

2. In Table 1, simulation power and formula power are quite confusing, and probably need a clearer naming or description in legends.

3. In the R package, the two functions are well-explained.

6. PLOS authors have the option to publish the peer review history of their article (what does this mean?). If published, this will include your full peer review and any attached files.

Reviewer #1: No

Reviewer #2: No

---

## [Author Response · Author response to Decision Letter 0]

16 Nov 2023

Comments from Reviewer 1

1- Have the authors made all data underlying the findings in their manuscript fully available?

Reviewer #1: No

Reviewer #2: Yes

Response: In the submission process, we provided the Github Link 'https://github.com/fangshuye/NMA-two-arms-sample-size' in the 'Data Availability' section. To ensure the reproducibility of all results presented in our manuscript, we have updated that Github repository to include all datasets and codes. This repository now covers 

• sample input data for R shiny app

• dataset and codes for simulation study

• code for generating relevant figures 

2 - This method is only tried on simulation analysis to compare the statistical power and standard error with existing network and without existing network. Have the authors ever applied this method on real clinical trial data by any chance to evaluate the power and standard error? I am considering simulation would be overfitting in some circumstances.

Response: 

In real clinical trial meta-analysis data, the truth (whether there is a difference and how large a difference) would be unknown. Statistical power and standard error can only be assessed under a situation when the truth is known. In a laboratory setting, it might be possible to set up an experiment with truth known. Yet it does not apply to meta-analysis of clinical trials.

We set up a simulation based on real-world data to the extent possible, allowing us to make meaningful comparisons and assess our methodology across a range of realistic scenarios. All simulation parameters are estimated from real data and scenarios are set within a reasonable range of sample sizes and effect sizes. 

3 - How did the authors select the existing network? Selecting an accurate and representative network from the existing options can be challengeable, given the potential for inaccuracies in available resources. Were there any specific protocols employed to navigate the incorporation of existing networks into this process?

Response: We chose the existing network from a previously published study on interventions for Bovine Respiratory Disease (BRD) in feedlot cattle. This selection was based on several considerations:

• The original paper employed rigorous protocols to ensure the accuracy and representativeness of the existing network. 

• We are familiar with the dataset and research questions involved.

• Our motivation for this current paper is to plan future trials related to this specific existing network, making it essential for our simulation study.

[1] O’Connor A, Yuan C, Cullen J, Coetzee J, Da Silva N, Wang C. A mixed treatment meta-analysis of antibiotic treatment options for bovine respiratory disease–an update. Preventive veterinary medicine. 2016;132:130–139

4 - How many existing network studies does it need at minimum to ensure a robust power of the new method? Will the statistical power of existing network meta-analysis affect the statistical power of the two-arm trial and sample size optimization?

Response: The number of existing network studies does not directly impact the power of the new method. The key determinant of the new method's power is the standard error of the estimated effect size between treatments A and B from existing Network Meta-Analysis (NMA) (Equation 6). In essence, a single existing study between A and B could suffice if it provides a reliable estimation with a small standard error.

The statistical power of the existing NMA, the statistical power of the two-arm trial analyzed with existing network, and sample size optimization, are directly influenced by the standard error of the estimated effect size from existing NMA. When this standard error is smaller, both the power of existing NMA and the power of the new method (combining new two-arm with existing network) increase, given a fixed total sample size. Alternatively, a smaller total sample size is needed to achieve a predefined level of power.

5 - Providing legends to elucidate your figures, such as in Figure 3, by explaining the significance of nodes and edges, would enhance clarity and comprehension.

Response: Thank you for your valuable suggestion. We have now added legends to Figure 3 to provide explanations for the significance of nodes and edges, following PLOS ONE figure guidelines. https://journals.plos.org/plosone/s/figures#loc-how-to-submit-figures-and-captions

6 - The clarity of Figure 4 and Figure 5 is compromised due to their low resolution. The figures don’t not show clearly to me. It would be better if you can improve it with higher resolution.

Response: We have improved their resolution by reducing the width when taking screenshots, resulting in larger and more legible text within Figures 4 and 5. We trust that this enhancement will provide better visibility.

Comments from Reviewer 2

1. The github page can have more example code shown in README.md

Response: Thank you for your suggestion. We have incorporated more example code into the README.md on our GitHub page for our published package (https://github.com/fangshuye/OssaNMA/tree/master#readme). For detailed instructions, we also recommend checking the vignettes of our package on CRAN (https://cran.r-project.org/web/packages/OssaNMA/vignettes/Introduction.html)

2. In Table 1, simulation power and formula power are quite confusing, and probably need a clearer naming or description in legends.

Response: Thank you for your feedback. We have added footnotes under Table 1 to provide explanations for "simulation power," "formula power," "simulation SE," and "formula SE." This clarification should enhance the understanding of the table.

3. In the R package, the two functions are well-explained.

Response: We appreciate your positive feedback regarding the clarity and explanation of the two functions in our R package.

---

## [Decision Letter · Decision Letter 1]

5 Dec 2023

OssaNMA: An R package for using information from network meta-analyses to optimize the power and sample allocation of a new two-arm trial

PONE-D-23-20293R1

Dear Dr. Wang,

We’re pleased to inform you that your manuscript has been judged scientifically suitable for publication and will be formally accepted for publication once it meets all outstanding technical requirements.

Kind regards,

Zechen Chong

Academic Editor

PLOS ONE

Additional Editor Comments (optional):

Reviewers' comments:

Reviewer's Responses to Questions

**Comments to the Author**

1. If the authors have adequately addressed your comments raised in a previous round of review and you feel that this manuscript is now acceptable for publication, you may indicate that here to bypass the “Comments to the Author” section, enter your conflict of interest statement in the “Confidential to Editor” section, and submit your "Accept" recommendation.

Reviewer #1: All comments have been addressed

Reviewer #2: All comments have been addressed

2. Is the manuscript technically sound, and do the data support the conclusions?

Reviewer #1: Yes

Reviewer #2: Yes

3. Has the statistical analysis been performed appropriately and rigorously? 

Reviewer #1: Yes

Reviewer #2: Yes

4. Have the authors made all data underlying the findings in their manuscript fully available?

Reviewer #1: Yes

Reviewer #2: Yes

5. Is the manuscript presented in an intelligible fashion and written in standard English?

Reviewer #1: Yes

Reviewer #2: Yes

6. Review Comments to the Author

Reviewer #1: Comments to OssaNMA: An R Package for using information from network meta-analyses to optimize the power and sample allocation of a new tow-arm trial.

The authors provided a comprehensive response addressing the concerns raised previously. While certain limitations from the study's design cannot be fully addressed, e.g. Only simulation analysis could be applied currently because authors explained that statistical power and standard error can only be assessed under a situation when the truth is known, and meta-analysis of clinical trials cannot be applied in the lab settings. However, I believe that publishing this work is still valuable to give information of evaluating power in the context of designing two-arm Randomized Controlled Trials (RCTs) for research purposes.

Reviewer #2: The study developed an R package and related Shiny app called OssaNMA for power and sample size optimization for a new two-arm trial. The tool was well-developed, and the Shiny app was easy to use, this could help researchers for future RCTs planning to get an optimal power and sample size. The findings and analysis presented in this study are beneficial to the research community.

The GitHub page has been updated from the previous recommendation. And all comments have been addressed.

7. PLOS authors have the option to publish the peer review history of their article (what does this mean?). If published, this will include your full peer review and any attached files.

Reviewer #1: No

Reviewer #2: No

---

## [Editor Report · Acceptance letter]

11 Dec 2023

PONE-D-23-20293R1 

OssaNMA: An R package for using information from network meta-analyses to optimize the power and sample allocation of a new two-arm trial 

Dear Dr. Wang:

I'm pleased to inform you that your manuscript has been deemed suitable for publication in PLOS ONE. Congratulations! Your manuscript is now with our production department. 

Kind regards, 

on behalf of

Dr. Zechen Chong 

Academic Editor

PLOS ONE